# Effect of Graphene Oxide on Mechanical Properties of Cement Mortar and its Strengthening Mechanism

**DOI:** 10.3390/ma12223753

**Published:** 2019-11-14

**Authors:** Yahui Wang, Jiawen Yang, Dong Ouyang

**Affiliations:** 1Research Center of Engineering Materials and Structural Durability, Jinan University, Guangzhou 510632, China; yahui520@stu2017.jnu.edu.cn (Y.W.);; 2School of Mechanics and Construction Engineering, Jinan University, Guangzhou 510632, China; 3China Panshi Museum of Concrete, Guizhou 551700, China; 4Guizhou Tongxuehui Technology Co. Ltd., Guizhou 551700, China

**Keywords:** graphene oxide, cement, mechanical properties, scanning electron microscopy (SEM), mechanism

## Abstract

The effects of the water–binder ratio and different graphene oxide (GO) sizes on the mechanical properties of GO-cement composites were systematically studied by preparing GO-cement mortars. The scanning electron microscopy observation (SEM) of the surface and fracture surface of cement pastes was carried out to study the morphology of cement hydration crystals in GO-cement systems under different space conditions. It was found that GO nanosheets significantly improved the compressive, flexural, and tensile strengths of cement mortars. When the dosage of GO nanosheets was 0.03% by weight of cement, the compressive, flexural, and tensile strengths at 28 days increased by 21.37%, 39.62%, and 53.77%, respectively, but GO was not found to be able to regulate the formation of flower-like cement hydration crystals. It was only shown that the growth space had an important influence on the morphology of hydrates. A possible working mechanism was proposed by which GO nanosheets prevented the expansion of microcracks in the cement pastes via a shield effect, thus enhancing the strength and toughness of the cement composites.

## 1. Introduction

Cement-based materials are regarded as one of the most important building materials due to their simple construction process, low-energy consumption, and the wide range of raw materials from which they can be synthesized. With the wide application of cement-based concrete, its defects, such as large self-weight, high brittleness, and low tensile strength, urgently need to be improved [1]. Micro-nano technology plays an increasingly important role in modern science and technology. It was found to be feasible to use nanomaterials to modify cement-based materials and improve mechanical properties and durability [2,3,4,5,6,7]. Graphene oxide (GO) nanosheets, which are a graphene derivative, have an extremely large, specific surface area, as well as excellent mechanical properties and thermal conductivity, and their surface contains a large amount of active oxygen-containing groups, such as hydroxyl, carboxyl, and epoxy groups [8]. It is easy to form GO-cement composites and disperse them into cement pastes due to their extremely strong hydrophilicity. At the same time, when compared with expensive fullerene and graphene, GO nanosheets are relatively cheap and raw graphite is easy to obtain. GO nanosheets have been applied widely in the field of nanocomposites, showing excellent modification properties and potential application value in polymer composites and inorganic composites [9,10,11,12,13,14]

In recent years, the use of GO as a reinforcing material for cement-based composites also attracted the attention of scholars, and some results have been published. Lv et al. [15] reported that the addition of 0.03% by weight of cement (bwoc) GO in cement mortars increased the tensile, flexural, and compressive strengths by 78.6%, 60.7%, and 38.9%, respectively. Pan et al. [16] reported that the use of GO (0.05% bwoc) increased the compressive and flexural strengths of GO-cement composites by 33% and 59% at 28 days. Lu et al. [17] reported that the addition of GO (0.08% bwoc) to strain-hardening cement-based materials increased the compressive and tensile strengths by 24.8% and 37.7%, respectively. Mokhtar et al. [18] reported that the addition of GO nanoplatelets (0.02% bwoc) to cement pastes increased the compressive strength by about 13%, and the addition of GO nanoplatelets (0.03% bwoc) to cement pastes increased the splitting tensile strength by 41%. Wu et al. [19] reported that the compressive strength of the concrete curing for 28 days increased from 12.84% to 34.08%, while the content of the GO nanosheets increased from 0.02% to 0.08% with a water–binder ratio of 0.5. The addition of GO nanosheets improved the flexural strength of concrete from 2.77% to 15.60% at 28 days when the content of GO nanosheets increased from 0.02% to 0.08%.

Studies have shown that graphene oxide nanosheets are very effective in enhancing the mechanical properties of cement-based materials due to their atom-thick two-dimensional structure [20]. However, the mechanism by which a very small amount of GO in cement-based composites is effective is still not fully understood. Some scholars carried out research on the hydration mechanism of a GO-cement system. Lin et al. [21] found that GO played a catalytic role in cement hydration. This was because the oxygen-containing functional groups of GO provided adsorption sites for water and cement, acted as nucleation sites for cement hydration products, and the water molecules adsorbed onto GO constituted reservoirs and water transport channels to further hydrate the cement. Lv et al. [22,23,24,25] used SEM to study the effect of GO on the microstructure of cement pastes and proposed the hydration mechanism of GO to control the morphology of cement hydrates, reporting that GO nanosheets exhibited the template effect and controlled cement hydrates (AFt, AFm, CH, C-S-H, etc.) to form flower-like crystals. Cui et al. [26] questioned the above regulation mechanism; they suggested that the flower-like crystals were not cement hydration products, but that a carbonization reaction occurred during specimen preparation, and that the crystals were calcium carbonate crystals (CaCO_3_). The growth space largely affects the morphology of cement hydrates, and the effect of the preparation process of the cement pastes is very large [27]. Therefore, when using SEM to analyze the morphology of GO-cement pastes, special attention should be paid to specimen preparation methods and processes in order to correctly understand the hydration mechanism of the GO-cement composites.

Some researchers reported that GO had no effect on the cement hydration process. For example, Horszczaruk et al. [28] reported that there was no significant difference in the morphology of cement hydration products after the incorporation of GO. It was found that the crystal phases were preserved upon the incorporation of GO into cement by XRD analysis, and the kinetics of the cement hydration process were not strongly affected by the incorporation of graphene oxide. Wang et al. [29] reported that GO did not promote hydration, but the incorporation of GO could affect the crystallization process and morphology of hydration products, especially calcium hydroxide. Pan et al. [16] reported the formation of strong interfacial forces due to chemical reactions between the carboxylic groups of GO and hydration products. Furthermore, the crack propagation trajectory on the fracture surface of GO-cement pastes was studied. It was reported that the main role of GO might be that GO effectively promoted the deflection of cracks in the pastes or forced the cracks to be inclined and twisted. Some scholars [30,31,32] also reported that GO inhibited the propagation of microcracks in cement-based materials. However, the research in this area seems to be somewhat inadequate; therefore, the strengthening mechanism of graphene oxide on cement-based materials will be the focus of this paper.

In this paper, the effects of the water-binder ratio and different GO sizes on the mechanical properties of GO-cement composites were systematically studied by preparing GO-cement mortars. Then, thermogravimetric analysis and pore structure tests were used to study the effect of GO on cement hydration. Moreover, through SEM observation of the surface and fracture surface of the cement pastes, as well as the fracture surface of the low-and high-water–binder-ratio cement pastes, the morphologies of the cement hydration products of the GO-cement system under different space conditions were studied. Previously-reported related studies were compared to explore the strengthening mechanism of GO on cement-based materials.

## 2. Experimental Procedures

### 2.1. Materials

The cement used in this study was Portland cement type II 42.5R produced by Guangda Cement Co., Ltd. (Huizhou, China). The chemical composition of the cement is listed in Table 1. The graphene oxide nanosheet dispersions used were purchased from Chengdu Institute of Chemistry, Chinese Academy of Sciences, with an oxygen content of 40%. They were synthesized by the modified Hummers method [33] and ultrasonically dispersed in deionized water. The size and thickness of the three kinds of GO produced were 0.5–3 μm and 0.55–1.2 nm, 8–15 μm and 1.75 nm, and >50 μm and 1.8 nm, respectively. The fine aggregate used in this study was natural river sand (Guangzhou, China), with a fineness modulus of 2.46. A polycarboxylate-based superplasticizer produced by Sika Building Materials Co., Ltd. (Guangzhou, China) was used in cement-based materials for workability purposes.

### 2.2. Characterization of GO

The crystal structure and interlayer distance of the GO were analyzed by a Brook D8 ADVANCE double-path powder X-ray diffractometer (Karlsruhe, Germany) with a 40 kV, 40 mA Cu radiation source. The molecular structure and functional groups of GO were characterized by a Nicolet Model 380 infrared spectroscopy analyzer (Waltham, MA, USA). The surface morphology of GO was observed using a JEM-2100F field-emission transmission electron microscope (FE-TEM) manufactured by JEOL (Tokyo, Japan) with an acceleration voltage of 200 kV. The graphene oxide solution (5.0 mg/mL) was diluted 1000 times, sonicated for 30 min, and two or three drops were dropped onto the copper mesh.

### 2.3. Preparation and Testing of Cement Pastes

The GO-cement pastes were composed of cement, water, and GO. The water–binder ratio was 0.55 or 0.2. The 450 g of cement and 247.5 g/90 g water and GO (0.03% by weight of cement with an average size of >50 μm) were mixed using standard curing for 28 days, and the control groups were plain cement pastes. SEM observation specimens were prepared using the surface method and the fracture surface method, and SEM fracture surface specimens were prepared with different water–binder ratios. The surface morphology with sufficient hydrate growth space and an appropriate internal morphology offering a limited growth space of the GO-cement pastes were studied.

The surface specimen preparation uniform stirred cement pastes being smeared onto cut 5 mm × 5 mm glass sheets with a glass rod, placed in a foam box, and covered with a moist towel to prevent water loss. The curing temperature was 20 ± 2 °C and the specimens were taken out after 3 days. The fracture surface specimen preparation involved uniform stirred cement pastes with a size of approximately 2 cm × 2 cm × 2 cm being prepared 28 days after being crushed. The specimens were kept in a standard curing room (relative humidity was ≥95%, temperature was 20 ± 2 °C) until SEM observation was performed.

A Phenom Pharos desktop scanning electron microscope, produced by Thermo Fisher Scientific (Waltham, CA, USA), was used to analyze the microscopic morphology of the specimens. The specimens were immersed in absolute ethanol and taken out for drying, and a platinum conductive layer (about 10 nm thick) was sprayed onto the specimens before the test.

In order to study the hydration reaction of cement pastes, the specimens were ground and subjected to quantitative thermogravimetric analysis (TGA). The samples were heated from 60 °C to 800 °C in an Ar atmosphere at a constant heating rate of 10 °C per minute by a synchronous thermal analyzer model STA449F1, produced by Germany NETZSCH Instrument Manufacturing Co., Ltd. (Selb, Germany). The Thermogravimetric-differential scanning calorimetry (TG-DSC) curves were plotted and the weight losses of the specimens were calculated. As different hydrates in the cement pastes corresponded to different decomposition weight losses, the degree of cement hydration could be qualitatively evaluated.

### 2.4. Preparation and Testing of Cement Mortars

The GO-cement composites consisted of cement, water, sand, and GO with a sand–binder ratio of 3. In this paper, the effects of different dosages of GO (0.00%, 0.01%, 0.03%, and 0.05% bwoc) on the mechanical properties of cement-based materials were investigated. Different water–binder ratios (W/B = 0.2, 0.3, and 0.4) were designed to study the effects of W/B on the mechanical properties of GO-cement composites. In order to decrease the effects of the addition of GO on fluidity [34], 1% to 2% (by weight of cement) polycarboxylic acid superplasticizer was added to the specimens to improve fluidity and GO could be uniformly dispersed in the cement pastes [31]. Few people have studied the effects of different GO sizes on cement-based materials. Therefore, in this study, graphene oxides with different sizes (0.5–3 μm, 8–15 μm, and >50 μm) were used to study the effects of the size of GO on the mechanical properties of cement mortars.

The GO dispersion solution, water, and superplasticizer required for preparing the GO-cement composites were uniformly mixed and stirred at a low speed for 30 s. Following this, the sand was added to the mixtures for 30 s, and the mortars were again stirred at a high speed for 30 s. After cessation of stirring for 90 s, the mixtures were stirred again at a high speed for 60 s to obtain GO-cement mortars. The mixed cement mortars were injected into the mortar test mold, slightly inserted, formed on the vibrating table, and smoothed. The specimens were kept in a standard curing room (relative humidity was ≥95%, temperature was 20 ± 2 °C) until the specified testing age was reached. According to the GB/T17671-1999 standard [35], the mixtures were molded into a cuboid of 40 mm × 40 mm × 160 mm to test the flexural and compressive strengths. The flexural strength was tested using an NYL-300 cement bending tester (Wuxi, China), and the compressive strength was tested on a KZY-500 cement compression tester (Shenyang, China) at a loading rate of 0.05 MPa/s.

The test mold used to determine the tensile strength of the mortars was 78 mm in length and 22.5 × 22.2 mm in the mid-section of the mortar eight-shaped mold, according to the ASTM C307 standard [36], as shown in Figure 1. Some scholars [37,38] reported that the eight-test method was easy to operate, had high efficiency, and fluctuation of the test data was small. In addition, the measured tensile strength was more realistic.

The pore structures of the cement mortars were measured by mercury intrusion porosimetry (MIP), and the porosity and pore size distribution of the cement mortars were tested using an Autopore IV 9500 automatic mercury instrument produced by Micromeritics Instrument Co. (Norcross, GA, USA).

## 3. Results and Discussion

### 3.1. XRD, FTIR, and TEM Analyses of GO

The X-ray diffraction (XRD) patterns of graphite and graphene oxide (GO) are plotted in Figure 2. The results indicated that the peak height of the graphite crystal was sharp and the peak appeared at 26.5° (2θ), whereas the peak appeared near 10° (2θ) for dry GO specimens and the interlayer distance of GO expanded to 0.859 nm from 0.341 nm for graphite. Graphite itself is very stable due to its properties and structure; therefore, there are large barriers to its direct participation in this reaction. After oxidation, the graphene interlayer distance increased to some extent, and the interactions between graphene layers weakened; therefore, GO could disperse into aqueous solutions easily.

The Fourier-transform infrared spectroscopy (FTIR) pattern of GO is shown in Figure 3. Five stretching vibration peaks appeared in the figure, which were at 854 cm^−1^, 1053 cm^−1^, 1617 cm^−1^, 1734 cm^−1^, and 3373 cm^−1^, respectively. The pattern recognized that these vibrational peaks were caused by the stretching vibration of epoxy groups (–O–), C–O, C–C/C=C, C = O, and –OH, respectively. The presence of these characteristic peaks indicated that, during preparation of GO, graphite was successfully oxidized due to the introduction of oxygen-containing groups, such as carboxyl groups, hydroxyl groups, epoxy groups and carbonyl groups; therefore, the GO nanosheets had good hydrophilicity.

The transmission electron micrograph (TEM) images of graphene oxide are shown in Figure 4. As can be seen from Figure 4, graphene oxide exhibited a tissue shape in a liquid state, and was wrinkled on the surface of the graphene oxide. The curl was very noticeable at the edge of the surface in particular. 

### 3.2. Effects of W/B on Mechanical Properties of GO-Cement System

The influences of the water–binder ratio on the compressive and flexural strengths of the GO-cement system at 28 days are shown in Figure 5. The strength increased with increasing GO contents until it reached 0.03% bwoc, followed by a decrease in strength with a further increase in GO content to 0.05% bwoc at 28 days. The compressive and flexural strengths of cement mortars increased with the decrease in the water–binder ratio.

Figure 5 shows that the highest compressive and flexural strengths among all of the specimens at 28 days increased by 10.97% and 25.45%, respectively, compared with the cement specimens without GO when the water–binder ratio was 0.4. When the water–binder ratio was 0.2 and the GO content was 0.03%, the compressive and flexural strengths reached the maximum value, which were 58.27 MPa and 14.8 MPa, which represented increases of 21.37% and 39.62%, respectively, compared with the control cement specimens. This indicated that adding a certain amount of GO could improve the strength of the cement mortars, especially the flexural strength. When the GO content was 0.05%, the compressive strength and the flexural strength started to decrease. As the amount of GO increased, the water demand increased significantly, which had a negative impact on the fluidity of the cement-based materials. At high GO dosages, the GO sheets tended to gather together due to intermolecular van der Waals forces, and the dispersion of the GO flakes in the cement pastes became difficult, thereby decreasing the strength of the cement-based composites. Comparing the strengths of GO-cement mortars under different water–binder ratios, the results showed that GO nanosheets had a better effect on cement-based materials when the water–binder ratio was low.

### 3.3. Effects of Different Diameters of GO on Mechanical Properties of the GO-Cement System

The effects of different diameters of GO on the compressive and flexural strengths of the GO-cement system at 28 days are shown in Figure 6. It can be seen from Figure 6 that the compressive strength of the cement mortar first increased and then decreased with the increase in graphene oxide content for three different diameters; the larger the diameter of GO, the higher the compressive and flexural strengths of the cement mortars. The optimal dosage of GO of three kinds of diameters (0.5–3 μm, 8–15 μm, and >50 μm) was 0.03% at 28 days. The highest compressive strengths of cement mortars corresponding to three kinds of diameters (0.5–3 μm, 8–15 μm, and >50 μm) were 51.71 MPa, 52.72 MPa, and 54.48 MPa, corresponding to increases of 14.45%, 16.68%, and 20.58%, respectively, compared with the control cement specimens, and the highest flexural strength increased by 11.59%, 20.31%, and 24.1%, respectively.

With the increase in the diameter of graphene oxide, the strengths of the cement mortars increased gradually, indicating that the large-diameter graphene oxide had a significant effect on the mechanical properties of cement mortars, especially regarding the improvement of the flexural strength. This phenomenon may be due to the fact that the large-diameter graphene oxide had a more obvious effect on the bridging and blocking of microcracks [39], which needs further study.

The effects of different dosages of GO (>50 μm) on the tensile strength of cement mortars are shown in Figure 7. It was observed that, as the amount of GO increased, the tensile strength of cement mortar first increased and then decreased. This trend was basically consistent with that seen regarding the compressive and flexural strengths. The highest strength among all of the specimens increased by 40.36% compared with that of the control cement specimens at the age of three days when the GO content was 0.01%. At the age of 28 days, cement mortars containing GO (0.03% bwoc) exhibited an approximately 53.77% increase in tensile strength compared with that of the control cement specimens.

### 3.4. Hydration Process of Cement Pastes

As can be seen from the TG-DSC curves at 28 days in Figure 8, the curves of all of the specimens showed three rapid weight losses. The first weight loss, located between 110 and 300 °C, was mainly due to the dehydration of C-S-H and ettringite. The second major weight loss, observed at 400–500 °C, corresponded to the dehydroxylation of calcium hydroxide (Ca(OH)_2_), another hydration product. The third weight loss occurred at about 700–800 °C, which was equivalent to decarbonization from the cement clinker, calcium carbonate [40]. Quantitative analysis of the first weight loss and the second weight loss was used as an indicator of the degree of hydration of the cement-based materials.

The weight losses of the cement pastes are summarized in Table 2. These values were calculated based on the test data presented in Figure 8. According to the quantitative analysis of the second stage, GO had no significant effect on the CH content generated during cement hydration. However, according to the weight loss in the first stage, it was determined that different amounts of GO had a slight influence on the C-S-H gel produced by cement hydration. Cement pastes containing GO (0.03% bwoc) exhibited about a 0.44% increase in the weight loss of the first stage compared with that of the control cement specimens, which may have been due to some water molecules adsorbed on GO, preventing the free water present in the gel pores from completely evaporating at 110 °C. According to the results of the thermogravimetric test, GO had no obvious promotion effect on the cement hydration process.

### 3.5. Pore Structures of Cement Mortars

The effects of the GO nanosheets on the pore structure of the cement mortars were investigated, and the results are shown in Table 3. It can be seen from Table 3 that the addition of graphene oxide reduced the average pore diameter of the cement mortars and increased the specific surface area and density of the cement mortars. Cement mortars containing GO (0.03% bwoc) exhibited about a 0.78% decrease in the porosity compared with that of the control cement specimens. These results indicated that the addition of GO nanosheets had no evident effect on the pore structures of the cement mortars.

According to the difference in pore sizes, the pores in the cement mortars were divided into gel pores (<10 nm), capillary pores (10–1000 nm), and macropores (>1000 nm), and the capillary pores were further divided into large pores (>100 nm) and small pores (10–100 nm) [41]. As shown in Figure 9, the incorporation of graphene oxide decreased the total pore volume of the cement mortars at 28 days, but the extent of the decrease was not significant. Graphene oxide nanosheets were added to reduce the large pores of the hardened cement mortars, so that the number of small pores was increased; the number of gel pores of cement mortars at 28 days did not significantly change. This may have been due to the fact that graphene oxide is a two-dimensional nanomaterial with a large layer, and graphene oxide nanosheets have a certain partition effect on large pores in hardened cement mortars, thereby separating some of the large pores into small pores [29].

### 3.6. SEM Morphology of GO-Cement Pastes

SEM images of a plain cement slurry surface after curing for three days at a water–binder ratio of 0.55 are shown in Figure 10. Observed at low magnification, the surface of the pastes mainly exhibited two morphology types, i.e., loose and dense zones. In the loose zone, accumulated bulk calcium hydroxide (CH) crystals were observed (Figure 10A). The CH crystal structure was complete and the size was large, about 10 μm. In the dense zone, a cluster of flower-like crystals was found (Figure 10B), and the size was about 0.5 μm. The elemental compositions of these microcrystals were analyzed by energy-dispersive spectroscopy (EDS), and it was confirmed that the crystals were CH, as seen in Table 4. It was observed that the influence of the growth space on the hydration products was very large. In an unrestricted space, the hydration products grew freely; conversely, the growth of hydration products was limited by space [27].

Figure 11 shows the SEM images of cement paste surfaces with 0.03% bwoc GO after curing for three days at a water–binder ratio of 0.55. Figure 11A shows that the morphology of the loose zone on the surface of the GO-cement pastes was exactly the same as that of the plain cement pastes, and accumulated CH crystals were also found. The morphology of the cement paste surfaces in the dense zone is shown in Figure 11B. It was clearly seen that there were fiber bundle-like crystals in the void between the cement particles, which were confirmed by EDX analysis to be CH, as seen in Table 4. It could be inferred from Figure 11B that the presence of a bundle of fiber-like CH crystals was due to the limited growth space.

In this paper, the surface morphology of cement pastes with GO was observed by SEM. Evidence that GO nanosheets regulated the morphology of CH, C-S-H, and AFt hydrates was not found, only that the growth space had an important influence on the hydrate morphology. Cement pastes are a complex heterogeneous system, and the growth of hydrates is bound to be limited by environmental space. The original fracture surface of the cement pastes more realistically reflected the growth state of the hydrates under the limited space available inside the pastes. Therefore, the SEM observation of the original fracture surface of the cement pastes was used to study the hydration mechanism of the GO-–cement pastes.

Figure 12 shows the fracture surface morphology of plain cement pastes after curing for 28 days at a water–binder ratio of 0.55. It was found that the different hydrates were intertwined with the surface morphology and that the morphology was complicated; therefore, it was difficult to identify a single hydrate crystal. A number of radial C-S-H crystals, needle-rod AFt crystals, and CH crystals were observed in the fracture surface microstructure of the plain cement paste (Figure 12B). It was more clearly seen in Figure 12C,D that the CH crystals grew from the center to the periphery to form flower-like crystals. This morphology was similar to the previously reported morphology of cement hydrated crystals [10,24]. The flower-like hydrated crystals were also observed in the cement pastes without GO and there were many pores inside the cement pastes with high water–binder ratios, thereby providing sufficient space for the growth of the hydrates.

Figure 13 shows the fracture surface morphology of the cement pastes with 0.03% bwoc GO after curing for 28 days at a water–binder ratio of 0.55. From Figure 13A–D, some flower-like crystals were observed again. At a low fraction of GO, it was quite challenging to identify the GO using SEM analysis due to its planar geometry and hydration products attached to the GO. Fortunately, GO was found in the GO-cement matrix, as shown in Figure 13E,F. The GO was inserted into the cement hydration products and was well-bonded to the cement matrix (Figure 13E). GO tended to bond to and tightly interlace with hydration products to prevent crack propagation under external loads. Good interfacial interactions between the GO and cement matrixes effectively transferred stress and improved the mechanical properties of the GO-cement composites. From Figure 13F, GO may have absorbed divalent calcium cations to form aggregates, wrinkles, and folds on the surface, thereby preventing crack propagation. GO aggregates behaved like fibers to improve the crack resistance, toughness, and intensity of the cement matrix, which was consistent with the results of Li [39,42].

Figure 14 and Figure 15 show the fracture surface morphology of the cement pastes after curing for 28 days at a water–binder ratio of 0.2. A cluster of fibrous-like crystals emerged in Figure 14A, which were confirmed to be C-S-H via EDX analysis, as seen in Table 4, which were very different from acicular AFt (Figure 12A). Figure 14B shows the local amplification of Figure 14A; these fibrous-like crystals improved the bending strength by preventing the propagation of cracks. The fracture surface images of the cement pastes with 0.03% bwoc GO are shown in Figure 15; the fibrous-like crystals were again found. After detailed study, the size of the fibrous-like crystals was in the range of 20–40 nm. In summary, by observing the original fracture surface morphology of cement pastes before and after adding GO at high and low water–binder ratios, there was no clear evidence that the presence of GO affected the morphology of hydrates, as reported previously [23,24,25,26]; these results only show that the effect of the growth space on the morphology of the hydration products is valid.

Combined with the above research, this paper believes that GO had no obvious promoting effect on the cement hydration process, and it was difficult to improve the strength of cement-based materials by regulating the growth of cement hydration products. The reason why it achieved the enhanced effect was due to the other important effects of GO on cement pastes. GO, a two-dimensional nanomaterial, has a larger specific surface area than carbon nanotubes and contains a large amount of oxygen-containing functional groups with good hydrophilicity, therefore it can be better dispersed in cement pastes. Microcracks that expanded in cement-based materials were more likely to encounter GO, so the probability of preventing microcrack propagation was greater, hereafter referred to as the cracking probability. Rafiee et al. [30] also reported that a possible toughening mechanism for GO in nanocomposites was crack deflection, which is the process by which an initial crack tilts and twists when it encounters a rigid inclusion. In addition, GO can be firmly anchored on the surface of the cement hydration products [20], which is attributed to the reaction between the CH, C-S-H crystals, and oxygen functional groups on the surface of GO [16,43]. Due to the close combination of GO and the surrounding cement pastes, the interface load transfer between the GO sheets and the cement pastes was enhanced, and the shield effect of preventing crack propagation of GO in the matrix may also have been enhanced. The larger specific surface area and higher aspect ratio allowed GO to act as a “shield” to better control the propagation of nanoscale cracks. When the microcracks propagated due to external loads, GO or GO aggregates formed a barrier that blocked the propagation of cracks, which increased the energy consumption required for crack growth. In this case, the crack was deflected or twisted instead of propagating in a straight manner, as shown in Figure 13F. Zhao et al. [32] reported that GO nanosheets provided a higher resistance to crack propagation, causing crack branching or deflection. The schematic diagram of the shield mechanism of GO in the GO-cement sample is illustrated in Figure 16. Esmaeeli et al. [44] reported that the early-age tensile stiffness and strength development of cementitious composites, such as mortar and concrete, were examined using a finite element model. Future study is needed to model the initiation and propagation of cracks in GO-cement composites using a mesomechanical model.

## 4. Conclusions

From the results of this study, the following conclusions can be drawn:Adding GO improved the compressive, flexural, and tensile strengths of cement mortars, with the increase in tensile strength being greater than that of the flexural and compressive strengths. The compressive, flexural, and tensile strengths of cement mortars incorporating 0.03% by weight of cement GO after curing for 28 days increased by 21.37%, 39.62%, and 53.77%, respectively, compared to that of mortars with GO. Comparing the effect of the different water–binder ratios and different sizes of GO on the mechanical properties of cement mortars, it was seen that the mechanical properties were more improved in the case of a low water–binder ratio or incorporation of large-diameter GO.According to the TGA results, GO has no significant effects on the Ca(OH)_2_ content generated during cement hydration at 28 days. Cement mortars containing GO (0.03% bwoc) exhibited about a 0.78% decrease in porosity compared with that of the control cement specimens by MIP testing. It was indicated that the incorporation of GO had no significant effect on the hydration process of cement.The surface and fracture surface morphology of the cement pastes with GO were almost the same as that of the plain cement pastes. It was easy to observe the flower-like hydrates from the surface of cement pastes with GO and plain cement pastes by SEM, and the flower-like crystals could be seen on the fracture surface of the plain cement pastes at a high water–binder ratio. GO was not found to regulate the morphology of crystals, which only indicated that the growth space had an important influence on the morphology of the hydrates.GO nanosheets have a high probability of preventing microcrack propagation; the mechanism of GO on cement-based composites to achieve this is mainly to prevent the expansion of microcracks via the shield effect to achieve strengthening and toughening. There are few studies regarding the use of GO to prepare high-performance concrete, which also calls for further research.

## Figures and Tables

**Figure 1 materials-12-03753-f001:**
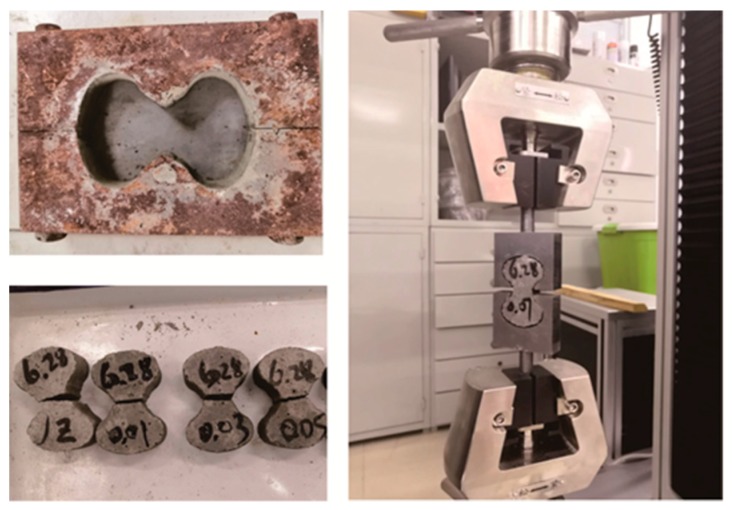
Eight-shaped mold and test method for tensile strength.

**Figure 2 materials-12-03753-f002:**
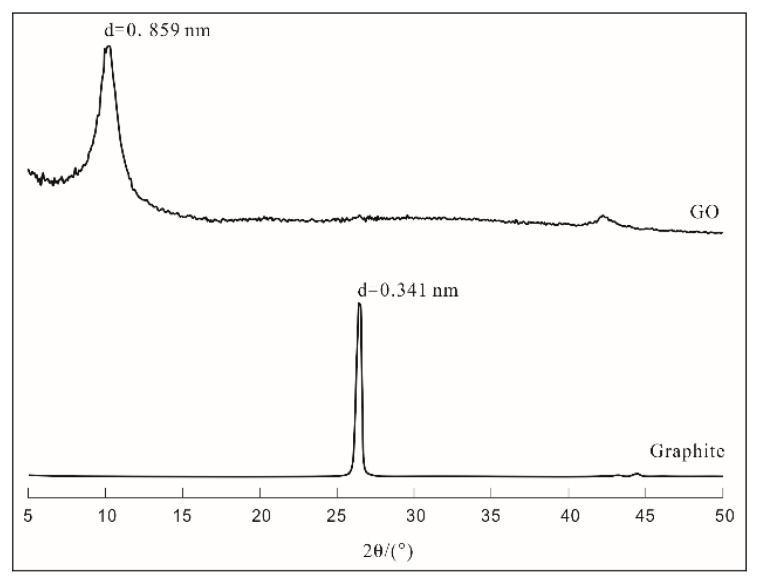
XRD patterns of graphene oxide (GO) and graphite.

**Figure 3 materials-12-03753-f003:**
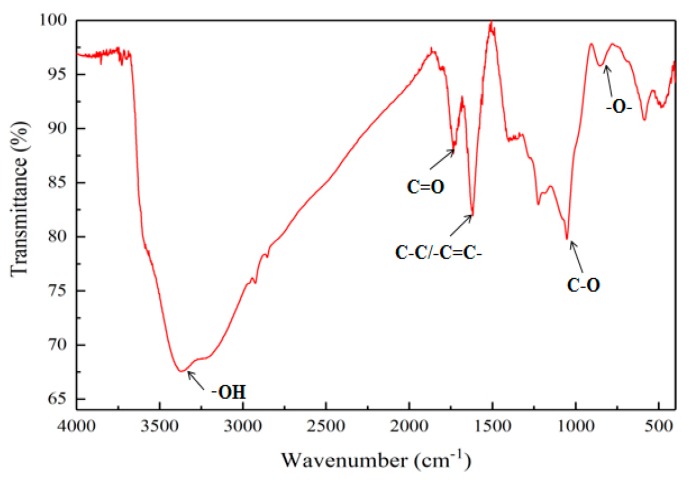
FTIR pattern of GO.

**Figure 4 materials-12-03753-f004:**
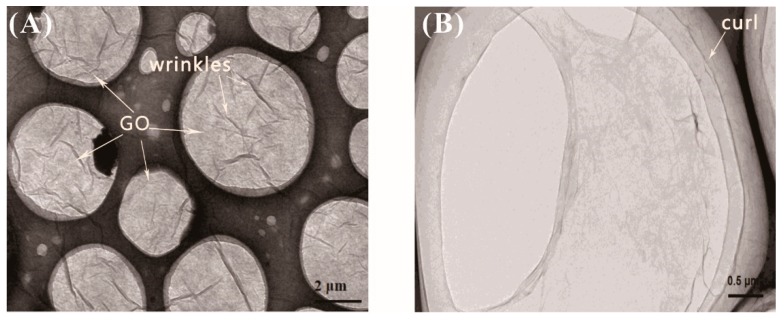
TEM images of GO (**A**) at ×20,000 magnification; (**B**) at ×50,000 magnification.

**Figure 5 materials-12-03753-f005:**
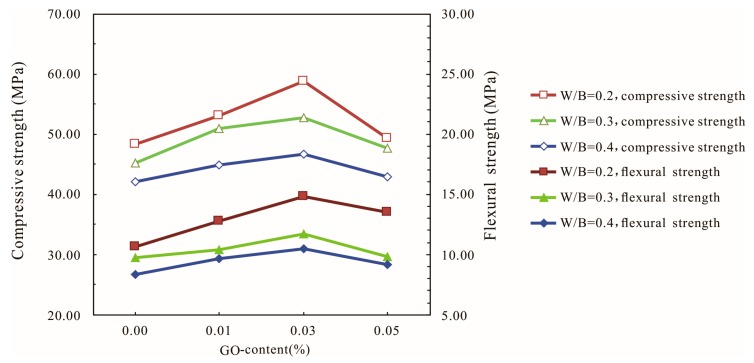
Influences of the water–binder ratio (W/B) on the compressive and flexural strength of the GO-cement system (cement composites were prepared by mixing 450 g cement, 1350 g sand, 90 g/135 g/180 g water, and different amounts of GO by standard curing for 28 days).

**Figure 6 materials-12-03753-f006:**
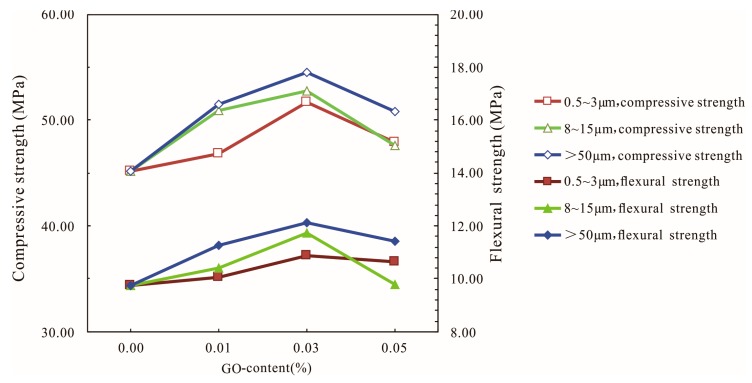
Influences of different diameters of GO on the compressive and flexural strengths of the GO-cement system (cement composites were prepared by mixing 450 g cement, 1350 g sand, 135 g water, and different amounts of GO (at three different sizes of 0.5–3 μm, 8–15 μm, and >50 μm) using standard curing for 28 days).

**Figure 7 materials-12-03753-f007:**
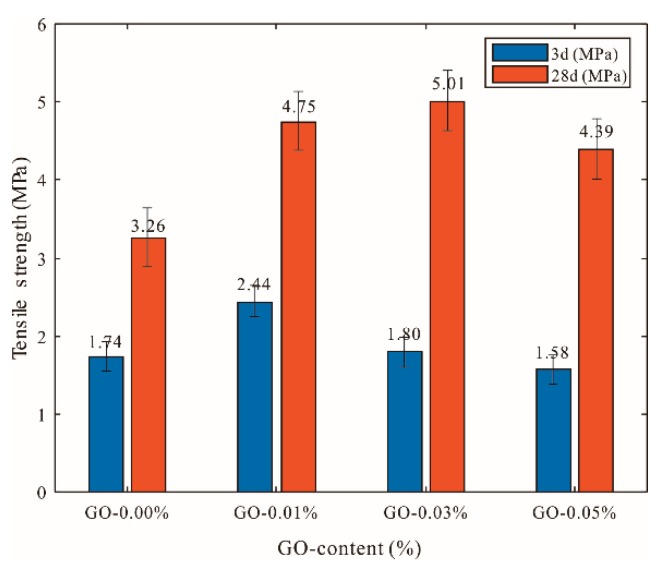
Influences of different GO dosages on tensile strength of GO-cement system (cement composites were prepared by mixing 450 g cement, 1350 g sand, 247.5 g water, and different amounts of GO (size of >50 μm) using standard curing for 3 and 28 days).

**Figure 8 materials-12-03753-f008:**
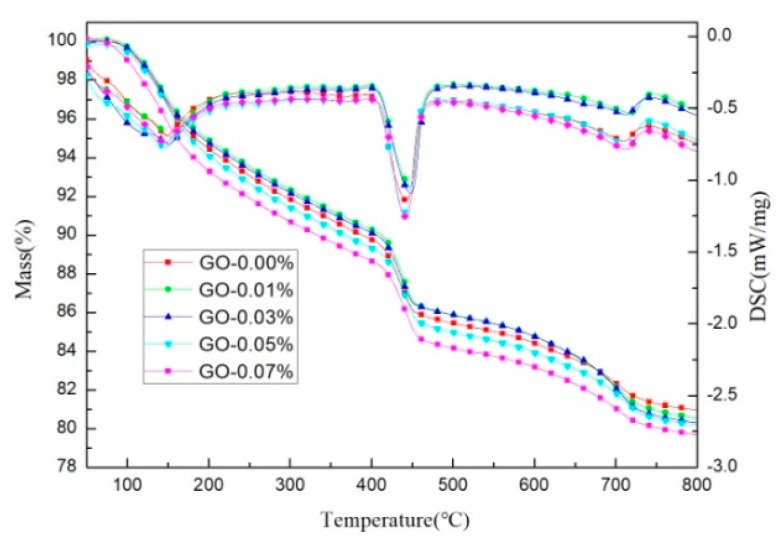
TG-DSC pattern of cement pastes with different dosages of GO (cement composites were prepared by mixing 450 g cement, 247.5 g water, and different amounts of GO (size of >50 μm) using standard curing for 28 days).

**Figure 9 materials-12-03753-f009:**
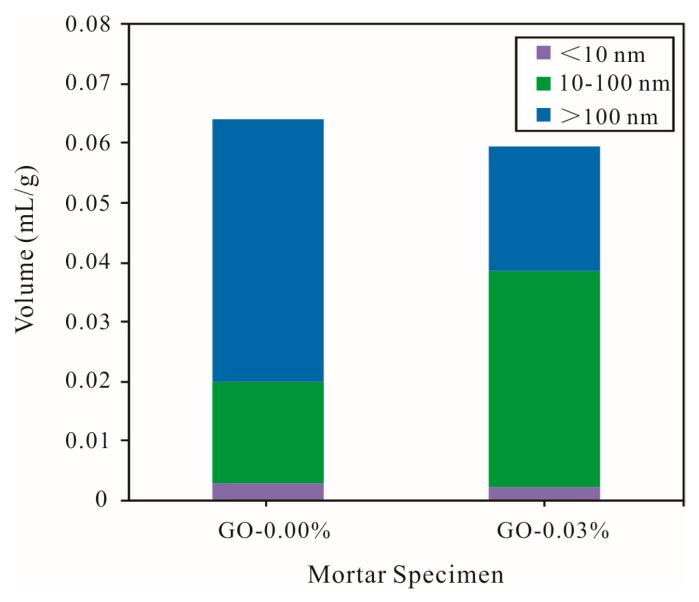
Pore volume distribution of different types of cement mortars (vement composites were prepared by mixing 450 g cement, 1350 g sand, 247.5 g water, and different amounts of GO (size of >50 μm) using standard curing for 28 days).

**Figure 10 materials-12-03753-f010:**
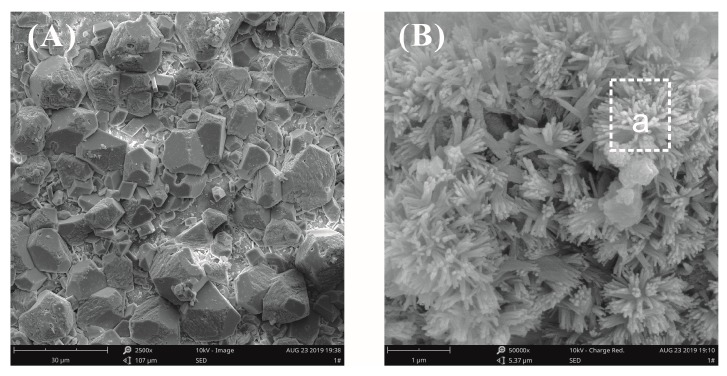
SEM images of plain cement paste surfaces at a water–binder ratio of 0.55. (**A**) Hydration morphology in the loose zone; (**B**) hydration morphology in the dense zone.

**Figure 11 materials-12-03753-f011:**
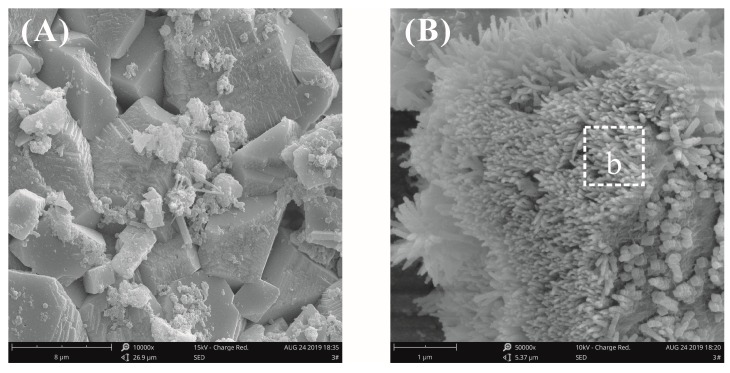
SEM images of the cement paste surfaces with 0.03% bwoc GO at a water–binder ratio of 0.55. (**A**) Hydration morphology in the loose zone; (**B**) hydration morphology in the dense zone.

**Figure 12 materials-12-03753-f012:**
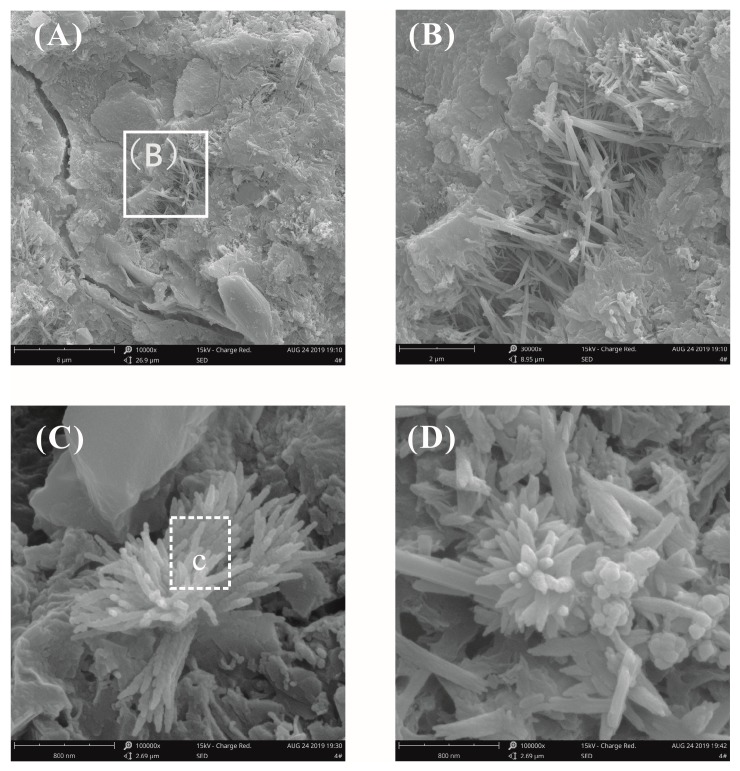
SEM images of the fracture surface of plain cement pastes at a water–binder ratio of 0.55. (**A**) Typical hydration morphology at low magnification; (**B**) local amplification for the area in the white box marked in image (**A**); (**C**,**D**) some flower-like crystals were observed.

**Figure 13 materials-12-03753-f013:**
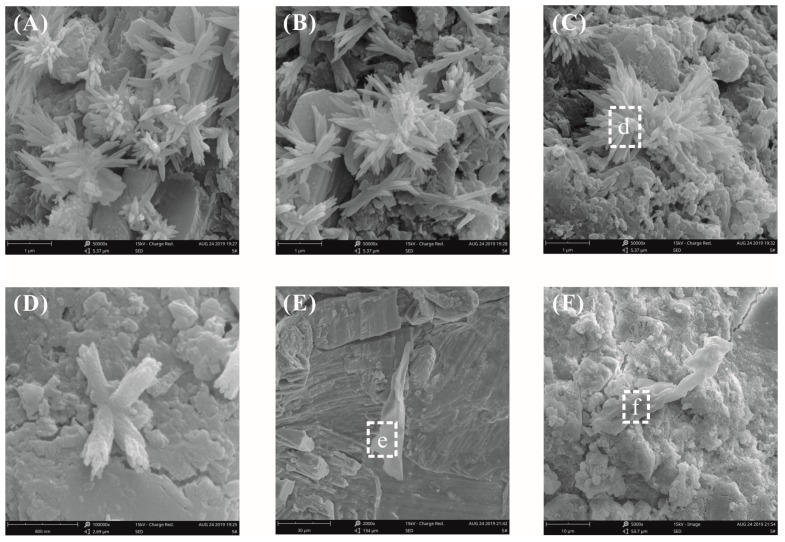
SEM images of the fracture surface of plain cement pastes with 0.03% bwoc GO at a water–binder ratio of 0.55. (**A**–**D**) Some flower-like crystals were found; (**E**,**F**) the GO pieces were found in the GO-cement matrix.

**Figure 14 materials-12-03753-f014:**
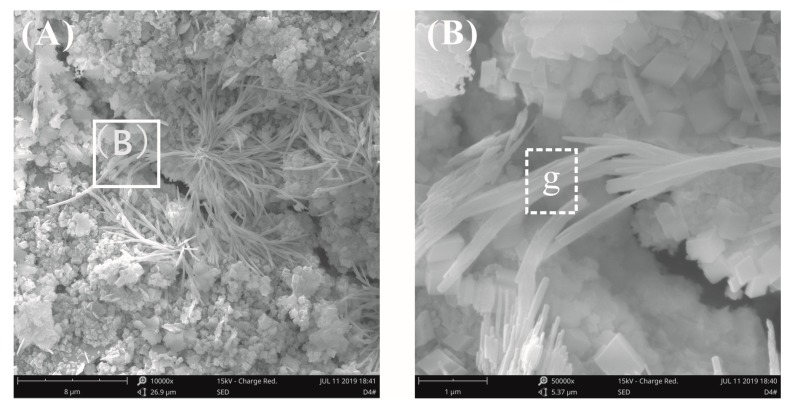
SEM images of the fracture surface of plain cement pastes at a water–binder ratio of 0.2. (**A**) A cluster of fibrous crystals was found. (**B**) Local amplification for the area in the white box marked in image (**A**).

**Figure 15 materials-12-03753-f015:**
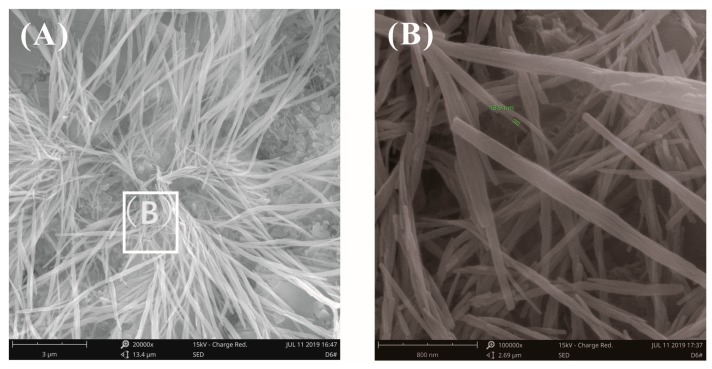
SEM images of fracture surface of cement pastes with 0.03% bwoc GO at a water–binder ratio of 0.2. (**A**) Fibrous-like crystals were found again. (**B**) Local amplification for the area in the white box marked in image (**A**).

**Figure 16 materials-12-03753-f016:**
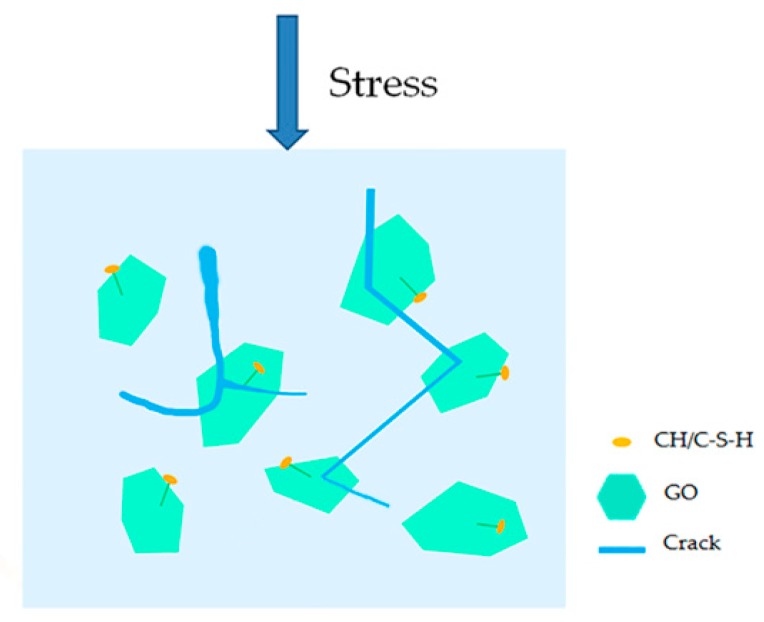
Schematic diagram of the shield mechanism of GO under an external load.

**Table 1 materials-12-03753-t001:** Chemical composition of the cement.

Material	Chemical Composition (wt%)						
	SiO_2_	CaO	Al_2_O_3_	Fe_2_O_3_	MgO	SO_3_	LOI
Cement	20.13	63.88	4.53	4.11	1.35	2.28	2.82

Note: LOI is the loss on ignition.

**Table 2 materials-12-03753-t002:** Weight loss of cement pastes at 28 days.

Paste Specimen	Weight Loss (%)		
	Stage 1	Stage 2	Stage 3
GO-0.00%	7.35	4.31	1.37
GO-0.01%	7.60	4.34	1.61
GO-0.03%	7.79	4.39	1.37
GO-0.05%	7.08	4.22	1.80
GO-0.07%	7.05	4.38	1.63

**Table 3 materials-12-03753-t003:** Pore structures of cement mortars at 28 days.

Mortar Specimen	Total Pore Volume (mL/g)	Specific Surface Area (m^2^/g)	Average Pore Size (nm)	Density (g/mL)	Total Porosity (%)	Pore Volume Ratio (%)		
						<10 nm	10–100 nm	>100 nm
GO-0.00%	0.0641	4.171	61.50	2.0742	13.30	4.88	26.52	68.64
GO-0.03%	0.0597	5.502	43.33	2.1002	12.52	4.36	60.30	35.18

**Table 4 materials-12-03753-t004:** Elemental composition of cement hydration crystals.

Place	No.	Element Percentage/wt%				Hydration Products
		Ca	Si	O	C	
Figure 10B	a	37.62	4.78	43.31	5.46	CH
Figure 11B	b	39.74	3.65	44.90	7.28	CH
Figure 12C	c	41.04	-	50.54	8.42	CH
Figure 13C	d	32.85	1.34	54.44	11.9	CH
Figure 13E	e	2.57	-	45.3	49.8	GO
Figure 13F	f	10.78	3.56	40.1	45.4	GO
Figure 14B	g	20.99	13.81	51.09	12.2	C-S-H

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
