# Peer review of "Effect of Graphene Oxide on Mechanical Properties of Cement Mortar and its Strengthening Mechanism"

_materials, 2019, doi:10.3390/ma12223753_

Round 1

Reviewer 1 Report

My review for this paper is in the attachment.

Author Response

Response to Reviewer 1 Comments

The anonymous Reviewer 1 provided thorough and critical reviews and suggestions. We went through all of the comments and suggestions carefully and made the changes one by one. The major modifications are shown as follows:

Point 1: Rewrite the title to be more grammatically correct and study on mechanism of what? It is unclear from this title (mechanism of hydration, mechanism of strength increase etc.?)

Response 1: Yes, we have rewrote the title such as “Effect of Graphene Oxide on Mechanical Properties of Cement Mortar and its Strengthening Mechanism” to more clear ones. We study mainly the strengthening mechanism of Graphene Oxide in cement based materials. Thanks for the comment and suggestion. Please see text, Line 2-3 of page 1.

Point 2: Abstract lines 21-22: please rewrite as it is unclear what you are trying to say: but the facts that GO can regulate flower like crystals what? Unclear and needs rewriting.

Response 2: Yes, we have rewrote the expression such as “but the fact that GO can regulate the formation of flower-like cement hydration crystals was not found” to more appropriate ones. Thanks for the comment and suggestion. Please see text, Line 17-18 of page 1.

Point 3: Lines 22-25: please rewrite that section e.g. ‘A possible working mechanism has been proposed and it is believed that GO nanosheets exhibited shielding effect and hence prevented the expansion of microcracks in cement paste. As a result enhanced strength and toughness of cement composited was noted.’

Response 3: Yes, we have rewrote the expression such as “A possible working mechanism was proposed by which GO nanosheets prevented the expansion of microcracks in the cement pastes by a shield effect, and thus enhanced the strength and toughness of the cement composites.” to  more appropriate ones. Thanks for the comment and suggestion. Please see text, Line 19-22 of page 1.

Point 4: Lines 30-371: Change…to…/insert…

Response 4: Yes, we have corrected these mistakes in new version. Thanks for the comment and suggestion.

Point 5: Lines 31: what do you mean by wide raw matrials? Clarify

Response 4: Yes, we have rewrote the expression such as “the wide range of raw materials from which they can be synthesized from” to more appropriate ones. Thanks for the comment and suggestion. Please see text, Line 28-29 of page 1.

Point 6: Lines 37: what type of performance? Please clarity...

Response 6: Yes, we have rewrote the expression such as “excellent mechanical properties and thermal conductivity” to more appropriate ones. Thanks for the comment and suggestion. Please see text, Line 34-35 of page 1.

Point 7: Lines 41-42: what do you mean. They have already been applied as indicated in the reference sources, right? Please clarify/rewrite.

Response 7: Yes, we have rewrote the expression such as “GO nanosheets have been applied widely in the field of nanocomposites, which have shown excellent modification properties and potential application value in polymer composites and inorganic composites” to more appropriate ones. Thanks for the comment and suggestion. Please see text, Line 39-41 of page 1.

Point 8: Lines 53-56: I do not understand this sentence. Did water to binder ratio increased GO weight % (seems bizarre) or when water to binder ratio was ate to 0.5 and the GO content increased from ... to ... wt% the compressive strength increased by 2.77, 15.60 respectively? Please rewrite/clarify.

Response 8: Yes, we have rewrote the expression such as “Wu et al. reported that the compressive strength of the concrete curing for 28 days was enhanced from 12.84% to 34.08% at the level of GO nanosheets from 0.02% to 0.08% for the water–binder ratio of 0.5. The addition of GO nanosheets improved the flexural strength of concrete in the range from 2.77% to 15.60% at 28 days when the content of GO nanosheets increased from 0.02% to 0.08%.” to more appropriate ones. Thanks for the comment and suggestion. Please see text, Line 51-55 of page 2.

Point 9: Lines 58-59: I do not understand. How does two dimensional structure benefit mechanical properties? in what way?

Response 9: Yes, we have corrected the expression such as “due to their atom-thick two-dimensional structure” to more appropriate ones. Graphene oxide nanosheets have an extremely large specific surface area and excellent mechanical properties and thermal conductivity due to their atom-thick two-dimensional structure. We studied mainly the strengthening mechanism of Graphene Oxide in cement based materials. Thanks for the comment and suggestion. Please see text, Line 57-58 of page 2.

Point 10: Lines 65-68: grammatically incorrect. Please rewrite this sentence.

Response 10: Yes, we have corrected the expression such as “Lv et al. [22–25] used SEM to study the effect of GO on the microstructure of cement pastes, and proposed the hydration mechanism of GO to control the morphology of cement hydrates, and reported that GO nanosheets exhibited the template effect and controlled cement hydrates (AFt, AFm, CH, C–S–H, etc.) to form flower-like crystals.” to more appropriate ones. Thanks for the comment and suggestion. Please see text, Line 64-67 of page 2.

Point 11: Flower like crystals of what? what mineral?

Response 11: Yes, the shape of the cement hydration crystals including ettringite (Ca6Al2(SO4)3)(OH)12⋅26H2O,AFt), Ca4Al2(OH)2⋅SO4⋅H2O, AFm), calcium hydroxide (Ca(OH)2, CH) and calcium silicate hydrate (3CaO⋅2SiO2⋅4H2O, C–S–H) gel is flower-like. Thanks for the comment and suggestion. Please see text, Line 66-67 of page 2.

Point 12: Grammatically incorrect. Please rewrite sentence between lines 68-71

Response 12: Yes, we have corrected the expression such as “Cui et al. questioned the above regulation mechanism; they considered that the flower-like crystals were not cement hydration products, but that a carbonization reaction occurred during specimen preparation, and that the nature of crystals was calcium carbonate crystals (CaCO3).” to more appropriate ones. Thanks for the comment and suggestion. Please see text, Line 67-70 of page 2.

Point 13: Please rewrite lines 78-81 to be grammatically correct. It was not preserved by XRD analysis right?

Response 13: Yes, we have corrected the expression such as “It was found that the crystal phases were preserved upon the incorporation of GO into the cement by XRD analysis and the kinetics of the cement hydration process were not strongly affected by the incorporation of graphene oxide.” to more appropriate ones. The crystal phases were preserved upon the incorporation of GO into the cement. Thanks for the comment and suggestion. Please see text, Line 77-80 of page 2.

Point 14: Rewrite lines 88-90 to be grammatically correct. It is unclear what you are trying to say.

Response 14: Yes, we have corrected the expression such as “However, the research in this area seems to be somewhat inadequate; therefore, the strengthening mechanism of graphene oxide on cement-based materials will be the focus of this paper.” to more appropriate ones. Thanks for the comment and suggestion. Please see text, Line 87-89 of page 2.

Point 15: rewrite lines 97-98 to be grammatically correct

Response 15: Yes, we have corrected the expression such as “The related studies reported in the past were compared to explore the strengthening mechanism of GO on cement-based materials.” to more appropriate ones. Thanks for the comment and suggestion. Please see text, Line 96-97 of page 2.

Point 16: Rewrite lines 103-105 as it in unclear and grammatically incorrect

Response 16: Yes, we have corrected the expression such as “The graphene oxide nanosheets dispersions used were purchased from Chengdu Institute of Chemistry, Chinese Academy of Sciences, with an oxygen content of 40%. They were synthesized by the modified Hummers method [30] and ultrasonically dispersed in deionized water.” to more appropriate ones. Thanks for the comment and suggestion. Please see text, Line 101-104 of page 3.

Point 17: Rewrite lines 126-127 as it in unclear and grammatically incorrect

Response 17: Yes, we have corrected the expression such as “The surface morphology with sufficient hydrate growth space and an appropriate internal morphology offering a limited growth space of the GO–cement pastes were studied.” to more appropriate ones. Thanks for the comment and suggestion. Please see text, Line 127-128 of page 3.

Point 18: Rewrite lines 137-138 as it in unclear and grammatically incorrect

Response 18: Yes, we have corrected the expression such as “The specimens were immersed in absolute ethanol, taken out for drying and a platinum conductive layer (about 10 nm thick) was sprayed onto the specimens before the test.” to more appropriate ones. Thanks for the comment and suggestion. Please see text, Line 137-139 of page 4.

Point 19: Rewrite lines 160-162 as it in unclear and grammatically incorrect

Response 19: Yes, we have corrected the expression such as “After stirring was stopped for 90 s, the mixtures were stirred at high speed for 60 s to obtain GO–cement mortars. The mixed cement mortars were injected into the mortar test mold, slightly inserted, and formed on the vibrating table and smoothed.” to more appropriate ones. Thanks for the comment and suggestion. Please see text, Line 161-163 of page 4.

Point 20: Rewrite lines 171-172 as it in unclear and grammatically incorrect

Response 20: Yes, we have corrected the expression such as “Some scholars reported that the eight-test method was easy to operate, had high efficiency, and fluctuation of the test data was small.” to more appropriate ones. Thanks for the comment and suggestion. Please see text, Line 171-172 of page 4.

Point 21: Rewrite lines 179-182 as it in unclear and grammatically incorrect

Response 21: Yes, we have corrected the expression such as “The results indicate that the peak height of the graphite crystal is sharp and the peak appears at 26.5° (2θ), the peak appears near 10° (2θ) for dry GO specimens, the interlayer distance of GO has expanded to 0.859 nm compared with that of graphite, 0.341 nm. Graphite itself is very stable due to its properties and structure; therefore, there are large barriers to its direct participation in the reaction.” to more appropriate ones. Thanks for the comment and suggestion. Please see text, Line 182-186 of page 5.

Point 22: Lines 196-197? What does it mean? I mean the last sentence....

Response 22: Yes, we have corrected the expression such as “Especially at the edge of its surface, the curl is very noticeable.” to more appropriate ones. Thanks for the comment and suggestion. Please see text, Line 201-202 of page 6.

Point 23: Rewrite lines 236-238 as it in unclear and grammatically incorrect

Response 23: Yes, we have corrected the expression such as “This phenomenon may be due to the fact that the large-diameter graphene oxide has a more obvious effect on the bridging and blocking of microcracks, which needs further study.” to more appropriate ones. Thanks for the comment and suggestion. Please see text, Line 246-247 of page 7.

Point 24: Rewrite lines 244-245 as it in unclear and grammatically incorrect

Response 24: Yes, we have corrected the expression such as “This trend is basically consistent with that of the compressive and flexural strength.” to more appropriate ones. Thanks for the comment and suggestion. Please see text, Line 249-250 of page 7.

Point 25: Rewrite lines 289-291 as it in unclear and grammatically incorrect

Response 25: Yes, we have corrected the expression such as “This may be due to the fact that graphene oxide is a two-dimensional nano-material with a large layer, graphene oxide nanosheets have a certain partition effect on the large pores in the hardened cement mortars, and separate a part of the large pores into small pores.” to more appropriate ones. Thanks for the comment and suggestion. Please see text, Line 297-299 of page 9.

Point 26: Rewrite lines 323-324 as it in unclear and grammatically incorrect

Response 26: Yes, we have corrected the expression such as “Evidence that GO nanosheets regulate the morphology of CH, C–S–H, and AFt hydrates was not found. It only shows that the growth space has an important influence on the hydrate morphology.” to more appropriate ones. Thanks for the comment and suggestion. Please see text, Line 328-329 of page 11.

Point 27: Rewrite lines 370-372 as it in unclear and grammatically incorrect

Response 27: Yes, we have corrected the expression such as “there is no clear evidence that the presence of GO affected the morphology of hydrates, as reported previously [23-26]; the results only explain that the effect of the growth space on the morphology of hydration products is valid.” to more appropriate ones. Thanks for the comment and suggestion. Please see text, Line 375-377 of page 13.

Point 28: Rewrite lines 388-390 as the sentence is unclear

Response 28: Yes, we have corrected the expression such as “When microcracks propagate because of external loads, GO or GO aggregates can form a barrier that blocks the propagation of cracks, which may increase the energy consumption required for crack growth.” to more appropriate ones. Thanks for the comment and suggestion. Please see text, Line 398-400 of page 13.

Reviewer 2 Report

In my opinion this paper is appropriate and it could be accepted for publication in the present form, although a general conclusion of this work, after specific conclusions, could be included.

Author Response

Thank you for arranging a timely review for our manuscript. We are pleased to know that our article is interesting for the scientific community in the field of civil engineering and it is appropriate to be accepted for publication in the present form.We have carefully evaluated the reviewers’ critical comments and thoughtful suggestions, responded to these suggestions point-by-point, and revised the manuscript accordingly. 

Thanks for your comment and affirmation, I have modified the article and live up to your expectations. Wish you the very best of luck in your job, every success in your endeavours, good health and a happy family.

Reviewer 3 Report

The manuscript by Wang et al., is an experimental work on the effect of graphene-oxides (GO) size, water/cement ratio on the performance of GO-cement composite. The author compare their effects by showing that GO can improve flexural, compressive and tensile performance by crack bridging method.

The paper is fairly well written and shows interesting results, but it need spelling and grammar correction. Please address the following:

Some specific comments to address:

Line 20, I” but the fact that GO can not regulate forming flower-like cement hydration crystals.” I think you mean despite the fact that. Line 42  “GO nanosheets are expected to be first applied in the field of

nanocomposites[9–14].” What does this mean?

Line 80 However, the research in this area seems to be somewhat inadequate,whether the enhanced modification effect of graphene oxide on cement-based materials is the shield blocking effect will be the focus of this paper.Rewrite this in two sentence please. Is there anyway to know the percentage of oxidation in GOs? Weigh percentage of oxygen to carbon? Can you add texts on Figure 4 and mention what each shape represents. the following work can help to explain strength graining due to having stronger filler in the cement. I think it helps to add as a reference.

Esmaeeli, Hadi S., et al. "A two-step multiscale model to predict early age strength development of cementitious composites considering competing fracture mechanisms." Construction and Building Materials 208 (2019): 577-600.

Author Response

Response to Reviewer 3 Comments

The anonymous Reviewer 3 provided thorough and critical reviews and suggestions. We went through all of the comments and suggestions carefully and made the changes one by one. The major modifications are shown as follows:

Point 1: Line 20:“but the fact that GO can not regulate forming flower-like cement hydration crystals.” I think you mean despite the fact that.

Response 1: Yes, we have rewrote the expression such as “but the fact that GO can regulate the formation of flower-like cement hydration crystals was not found” to more appropriate ones. Thanks for the comment and suggestion. Please see text, Line 17-18 of page 1.

Point 2: Line 42:“GO nanosheets are expected to be first applied in the field of

nanocomposites [9–14].” What does this mean?

Response 2: Yes, we have rewrote the expression such as “GO nanosheets have been applied widely in the field of nanocomposites, which have shown excellent modification properties and potential application value in polymer composites and inorganic composites [9–14]” to more appropriate ones. Thanks for the comment and suggestion. Please see text, Line 39-41 of page 1.

Point 3: Line 80: “However, the research in this area seems to be somewhat inadequate, whether the enhanced modification effect of graphene oxide on cement-based materials is the shield blocking effect will be the focus of this paper. Rewrite this in two sentence please. Is there anyway to know the percentage of oxidation in GOs? Weigh percentage of oxygen to carbon? Can you add texts on Figure 4 and mention what each shape represents. the following work can help to explain strength graining due to having stronger filler in the cement. I think it helps to add as a reference.

Esmaeeli, Hadi S., et al. "A two-step multiscale model to predict early age strength development of cementitious composites considering competing fracture mechanisms." Construction and Building Materials 208 (2019): 577-600.

Response 3: Yes, we have corrected the expression such as “However, the research in this area seems to be somewhat inadequate; therefore, the strengthening mechanism of graphene oxide on cement-based materials will be the focus of this paper.” to more appropriate ones. We have added more information about GO in new version. Please see text, Line 102-103 of page 3. We have changed the Fig. 4 in new version, and we add the literature you suggested as a reference. Thanks for the comment and suggestion.

Reviewer 4 Report

The article is interesting for the scientific community in the field of civil engineering. However, it must be seriously corrected so that it can be published.

I have comments:

Line 101 - wrong cement name. Should be: Portland cement.

The source of data given in Table 1 should be provided.

Fig. 2 is hardly legible.

Fig. 7 - unit not written correctly, must be MPa. Please enter for posts (deviation) - bars. Tensile strength testing often shows scatter of results, statistics are needed.

Except for item - 3.6 SEM Morphology of GO-Cement Pastes, the entire manuscript did not discuss the results of the research in relation to literature. It is rather a technical test report. A thorough analysis based on the research of other scientists is needed.

The manuscript is prepared a little chaotically. There are many editorial mistakes. A general improvement in manuscript editing is needed. The authors once use a space before units, other times words are "stuck" not only with the unit, e.g. lines 106, 114, 122, 136, 155, 169, 170 and others in all paper.

There is a different font in Table 3. 

The references do not include DOI numbers that are required in the Materials journal.

Author Response

Response to Reviewer 4 Comments

The anonymous Reviewer 4 provided thorough and critical reviews and suggestions. We went through all of the comments and suggestions carefully and made the changes one by one. The major modifications are shown as follows:

Point 1: Line 101 - wrong cement name.  Should be: Portland cement.

Response 1: Yes, we have corrected this mistake in the version. Thanks for the comment and suggestion. Please see text, Line 100 of page 3.

Point 2: The source of data given in Table 1 should be provided.

Response 2: Yes, we have done it. The chemical composition of the cement is provided by Guangda Cement Co., Ltd. (Huizhou, China) mentioned in the article. Thanks for the comment and suggestion. Please see text, Line 100-101 of page 3.

Point 3: Fig. 2 is hardly legible.

Response 3: Yes, we have changed the Fig. 2 in new version. Thanks for the comment and suggestion. Please see text, Line 189 of page 5.

Point 4: Fig. 7 - unit not written correctly, must be MPa. Please enter for posts (deviation) - bars. Tensile strength testing often shows scatter of results, statistics are needed.

Response 4: Yes, we have changed the Fig. 7 in new version. Thanks for the comment and suggestion. Please see text, Line 257-259 of page 8.

Point 5: Except for item - 3.6 SEM Morphology of GO-Cement Pastes, the entire manuscript did not discuss the results of the research in relation to literature. It is rather a technical test report. A thorough analysis based on the research of other scientists is needed.

Response 5: Yes, we have done it. We have added more thorough analysis based on the research of other scientists in new version. Thanks for the comment and suggestion. Please see text, Line 389-407 of page 13.

Point 6: The manuscript is prepared a little chaotically. There are many editorial mistakes. A general improvement in manuscript editing is needed. The authors once use a space before units, other times words are "stuck" not only with the unit, e.g. lines 106, 114, 122, 136, 155, 169, 170 and others in all paper.

Response 6: Yes, we have corrected these editorial mistake in the version. Thanks for the comment and suggestion.

Point 7: There is a different font in Table 3.

Response 7: Yes, we have changed the Table 3 in new version. Thanks for the comment and suggestion. Please see text, Line 289 of page 9.

Point 8: The references do not include DOI numbers that are required in the Materials journal.

Response 8: Yes, we have done it. We have added DOI numbers of references. Thanks for the comment and suggestion.

Round 2

Reviewer 4 Report

Thank you for the Authors' corrections. I recommend the article to be published.

Author Response

Dear Reviewer,

Thank you for agreeing to publish the article. We are pleased to know that our article is interesting for the scientific community in the field of civil engineering and it is appropriate to be accepted for publication in the present form. 

Thanks for your comment and affirmation, I have modified the article and live up to your expectations. Wish you the very best of luck in your job, every success in your endeavours, good health and a happy family.

Kind regards,

Mr. wang